# Gradient-Free De Novo Learning

**DOI:** 10.3390/e27090992

**Published:** 2025-09-22

**Authors:** Karl Friston, Thomas Parr, Conor Heins, Lancelot Da Costa, Tommaso Salvatori, Alexander Tschantz, Magnus Koudahl, Toon Van de Maele, Christopher Buckley, Tim Verbelen

**Affiliations:** 1Queen Square Institute of Neurology, University College London, London WC1E 6BT, UK; k.friston@ucl.ac.uk; 2VERSES Research Lab, Los Angeles, CA 90016, USA; conor.heins@verses.ai (C.H.); tommaso.salvatori@verses.ai (T.S.); alec.tschantz@verses.ai (A.T.); c-magnus.koudahl@verses.ai (M.K.); toon.vandemaele@verses.ai (T.V.d.M.); christopher.buckley@verses.ai (C.B.); tim.verbelen@verses.ai (T.V.); 3Nuffield Department of Clinical Neurosciences, University of Oxford, Oxford OX1 2JD, UK; thomas.parr@ndcn.ox.ac.uk; 4ELLIS Institute Tübingen, 72076 Tübingen, Germany; 5Department of Informatics, University of Sussex, Brighton BN1 9RH, UK

**Keywords:** active inference, active learning, Bayesian model selection, compression, structure learning, planning, induction

## Abstract

This technical note applies active inference to the problem of learning goal-directed behaviour from scratch, namely, de novo learning. By de novo learning, we mean discovering, directly from observations, the structure and parameters of a discrete generative model for sequential policy optimisation. Concretely, our procedure grows and then reduces a model until it discovers a pullback attractor over (generalised) states; this attracting set supplies paths of least action among goal states while avoiding costly states. The implicit efficiency rests upon reframing the learning problem through the lens of the free energy principle, under which it is sufficient to learn a generative model whose dynamics feature such an attracting set. For context, we briefly relate this perspective to value-based formulations (e.g., Bellman optimality) and then apply the active inference formulation to a small arcade game to illustrate de novo structure learning and ensuing agency.

## 1. Introduction

This technical note introduces an approach to sequential policy optimisation using active inference, learning and selection. Active inference is an application of the free energy principle that offers a complementary perspective on goal-directed behaviour and agency in natural and artificial systems [1]. In brief, the free energy principle allows one to describe any open system—i.e., a system that senses and acts upon its environment—as inferring the causes of its observations and consequent actions. Technically, the free energy principle can be regarded as an application of the maximum path entropy principle—also known as principle of maximum calibre—to measurement or observation [2,3]. Heuristically, it can be read as saying that any system that maintains itself within some characteristic states can be described as if it was inferring the latent causes of its observations—via minimising variational free energy—and realising states that are characteristic of the kind of thing it is—via minimising expected free energy. This somewhat tautological description of self-organisation can be leveraged practically by specifying preferred or characteristic states as priors in a generative model. One then invokes standard (active) inference schemes to minimise variational and expected free energy to evince perception and action, respectively. However, to simulate, reproduce or realise agentic behaviour, one has to specify a generative model of an agent’s exchange with the world. What follows is one approach to learning such models in the absence of any prior specification; in other words, learning—de novo—the structure and parameters of a generative model from, and only from, observations or data.

### 1.1. Agents and Attractors

First principles accounts of self-organisation often call upon variational principles of least action. For example, in the path integral formulation of the free energy principle [2], Bayes-optimal behaviour is an emergent property of certain random dynamical systems [4,5] that can be distinguished from their environment (via a Markov blanket or set of inputs and outputs). In this setting, optimal behaviours are paths of least action on a statistical manifold with a well-defined geometry (i.e., Fisher information metric) and Lagrangian (i.e., variational free energy). This description applies most naturally to joint (agentic) systems with a pullback attractor [5,6], whose attracting states that are characteristic of the agent in question. Crucially, this means the agent revisits (the neighbourhood of) states that it previously occupied, leading to a description of self-organisation as an itinerant flow or orbit on an attracting manifold. Note that there are important classes of open systems (e.g., certain reaction–diffusion systems) where comparable variational descriptions have not been established [7]; our focus is on agents amenable to the free-energy formulation.

By the Helmholtz-Hodge decomposition, this flow has dissipative and conservative parts [8,9,10]. In other words, the flow or dynamics can be separated into a dissipative (curl-free gradient flow) and a conservative (divergence-free solenoidal flow) part that circulates on the isocontours of variational free energy. When pursuing paths of least action, the conservative dynamics predominate, as in classical mechanics, biorhythms, lifecycles or, on an evolutionary timescale, Red Queen dynamics [11,12]. There are many instances of this sort of dynamics; ranging from generalised Lotka-Volterra models of winnerless competition [13] through to heteroclinic cycles that have been used to describe autonomous behaviour and neuronal dynamics [14,15]. Note that for highly non-smooth, multiscale dynamics (e.g., strong turbulence, certain discrete Markov systems), such separations may be problematic or only approximate; our use is heuristic.

Heteroclinic cycles are of particular interest here because they rest upon visiting a sequence of unstable fixed points. Any system pursuing paths of least action on a heteroclinic cycle will look as if it is goal-directed, seeking out successive fixed points. If we associate unstable fixed points with rewards, it will look as if the system (i.e., agent) is maximising the number of sparse rewards it expects to encounter. On this view, one can see a close connection between expected value and attractors, defined in the sense of the free energy principle [16]. Does this mean that one can replace the free energy principle with the Bellman optimality principle [17,18,19]?

Not quite. The Bellman optimality principle—and accompanying reinforcement learning schemes such as Q-learning and its deep-network instantiations [20,21]—seeks to optimise a value (i.e., reward) function of the states (i.e., observations and actions) of an agent. However, the agent’s (internal) states are belief states because they have been shown to evolve on a statistical manifold, where each point on the manifold corresponds to a probability density or (Bayesian) belief [22,23,24,25,26]. This means that movement or flow on the statistical manifold corresponds to Bayesian belief updating or inference; hence, active inference [25]. Technically, this suggests that the objective function that underwrites inference and learning is not a function of *states* but a functional of *beliefs about states*. This functional is the (variational and expected) free energy of belief states (i.e., Bayesian posteriors). This suggests that Bellman optimality is, by construction, suboptimal from the point of view of Bayes optimality: as exemplified by sample inefficiency, susceptibility to local minima, generalisation failures, and so on, e.g., [27]. For further comparison of reinforcement learning and active inference, please see [18,28,29]. On this view, it may be possible to learn goal-directed behaviour more efficiently than value-based reinforcement learning (e.g., tabular or deep Q-learning) through a free energy formulation. So, how could one apply variational principles of least action to realise that efficiency?

### 1.2. Model Selection

Under active inference, learning reduces to finding the structure and parameters of a generative model that minimise expected free energy. After this model has been discovered, one can apply standard belief updating and learning schemes to realise optimal behaviour in the dual sense of Bayesian decision theory [30] and optimal Bayesian design [31,32], i.e., to maximise expected value and information gain under the generative model selected. One could argue there are two basic approaches to model selection. The first is a top-down approach, in which one starts with an overly expressive (i.e., parameterised) model and reduces model complexity in light of empirical data or observations. In statistics, this is known as Bayesian model reduction [33,34]. In this setting, the agenda is to minimise model complexity, noting that (negative) variational free energy is a lower bound on log marginal likelihood or model evidence (ELBO), and log evidence is accuracy minus complexity [35]. Most machine learning schemes adopt this top-down approach, where Bayesian model reduction is replaced by dropout, pruning and various forms of regularisation, e.g., [36]. Here, neural networks play the role of a full model that is subsequently reduced or pruned to fit the application at hand. The alternative is a bottom-up approach, where one grows the model to explain successive data encountered, e.g., [37,38]. This leads to a fast kind of structure learning—e.g., [39]—in which the model is equipped with new parameters and dependencies when, and only when, a new observation cannot be explained in terms of the (latent) causes of past observations.

It is this bottom-up approach we pursue in the current paper, where model complexity approaches its Bayes-optimal level from below, as opposed to Bayesian model reduction in which the optimal complexity is approached from above. These two approaches have distinct offerings: the top-down approach offers a generalist architecture that can be specialised (i.e., reduced) to adapt to a particular application. Conversely, the bottom-up or de novo approach starts with nothing to furnish a specialist agent that is apt for the situation at hand. Figure 1 illustrates the distinction schematically. For clarity, here “top-down” and “bottom-up” refer to model-selection strategies (pruning an initially expressive model versus growing a minimal model to sufficiency), not to selecting the right spatial or temporal scale.

### 1.3. Rewards and Punishment

The objective of active inference is not to maximise reward or minimise cost. It is to maximise the evidence for a generative model of rewarded behaviour, under cost constraints. Treating costs as implementing constraints speaks to a qualitative distinction between rewards and punishments, which does not feature in reinforcement learning or behavioural economics. Classical (non-Bayesian) behaviourist or reinforcement learning accounts usually treat punishment as negative reward and then characterise asymmetries in the sensitivity to positive and negative rewards, e.g., [43,44,45,46]. However, from the perspective of specifying an attracting set, rewards and punishments play qualitatively distinct roles. Rewards are typically sparse and specify regimes of state-space an agent is likely to revisit (e.g., unstable fixed points in a heteroclinic cycle). In contrast, punishment-as-constraint specifies regimes of state-space an agent is never found in, i.e., that would be very surprising for—or uncharacteristic of—the agent in question. The key difference—between this reading of punishment and reward—is that punishment is avoided at all times, whereas reward can only be encountered in the future. Another way of looking at this is that there are many more ways of being dead (punished) than alive (rewarded) [47].

The distinction between punishment and rewards can be understood in terms of reaching intended (i.e., goal) states that generate rewarding outcomes, while circumnavigating costly states that would generate aversive outcomes. This distinction becomes operationally acute in sequential policy optimisation: in this illustrative work, we use inductive inference to realise goal-directed behaviour. Inductive inference leverages precise predictive posteriors to implement a simple form of backwards induction [48]. Specifically, state transitions are rolled out backwards in time to identify the smallest number of steps from the current (inferred) state to any goal or intended state. This allows subsequent states to be removed from policy selection if they do not afford a path to goal. By iterating this process at every time step, the agent pursues the shortest path to the nearest goal, i.e., a path of least action. However, for this kind of active inference to work, one needs precise beliefs over discrete states, which mandates a discrete generative model, e.g., a partially observed Markov decision process (POMDP) model. In this setting, constraints on paths are implemented by precluding transitions to costly (punished) states, prior to computing the shortest path to intended (rewarded) states. In short, punishment furnishes constraints on—or contextualises—goal-directed behaviour. Once a punishment is observed, its cost is incorporated and subsequent pruning precludes those transitions, so constraints immediately reshape the attracting set, whereas rewards remain prospective and define intended goal states for backward induction. Interestingly, we will see that an efficient model of goal-directed behaviour precludes transitions to costly states.

In moving from continuous random dynamical systems to their discrete homologues, random or stochastic differential equations—and associated Fokker Planck equations—are replaced by probability transition tensors and associated Master equations [10,49,50]. In some senses, the attendant maths—of discrete time systems—become easier to implement and analyse, e.g., by appeal to topological mixing and transitivity [51]. We will call on some of these advantages in the setting of POMDPs. Technically speaking, de novo learning is a procedure for discovering a set of states with *topological transitivity* [52], such that the forward orbits from certain (goal) states have a non-empty intersection with the backward orbits of goal states, furnishing a closed orbit through goal states. In this setting, goal states become *hypercyclic* points. In what follows, we will associate forward and backward orbits with basins of attraction, which are effectively enlarged until a—possibly dense—closed orbit emerges.

One noteworthy aspect of discrete inference and learning schemes is that they admit gradient free updates; furthermore, these updates can be made fully Bayesian by equipping their parameters with simple conjugate priors. In other words, unlike learning nonlinear continuous state space models—e.g., with backpropagation of errors—learning under discrete models with simple conjugate priors allows the use of fixed point iterations [53,54], also known as coordinate descent, e.g., [55]. The title of this paper is a nod to this gradient free aspect. However, it also foregrounds a more fundamental aspect of optimal behaviour: if optimal behaviour corresponds to taking paths of least action, then the ensuing dynamics are conservative and divergence-free. If the dynamics are divergence-free, gradient flows are precluded. This means that an agent with the right generative model will look as if it is pursuing orbits. (Orbits in this paper are read as complete orbits in random dynamical systems; namely, an orbit that exists both forwards and backwards in time. In discrete systems, this is the union of forward and backward orbits). We are interested in the closed orbits, such that every point of the orbit evolves to itself under some group action (e.g., evolution function or transition). See [56] for a topological example on its statistical manifold, with no fluctuations in variational free energy. We will see numerical examples of ‘surfing uncertainty’ [57] in this gradient-free fashion later.

### 1.4. Overview

In the following, we first rehearse the basics of active inference, learning and selection; namely, free energy minimising processes that implement belief-updating about the latent causes, parameters and structure of a generative model, respectively. This paper focuses on discrete state space models, where the (Bayesian) mechanics of belief-updating can be reduced to straightforward tensor operations. However, because we will be seeking paths of least action, we require generalised (POMDP) models in which generalised states generate *paths* at a lower level [39]. Furthermore, to deal with high dimensional observations, we will use a particular—i.e., renormalising—form of model that allows for scale-free inference and learning [58]. The subsequent section focuses on the fast structure learning of pullback attractors that are constrained by rewards and punishments. This rests upon selectively retaining transitions, among generalised states, that underwrite closed orbits. The Section 4 illustrates the application of these procedures to a small arcade game, to illustrate the various stages of learning and inference. We conclude with a brief discussion of the limitations and generalisations of this scheme. (To avoid overburdening the main text, some technical details have been placed in figure legends).

## 2. Active Inference, Learning and Selection

This section overviews the model used in subsequent sections. This model generalises a partially observed Markov decision process (POMDP) by equipping it with random variables called *paths* that specify transitions among latent states. These models are designed to be composed hierarchically, in a way that introduces a separation of temporal scales.

### 2.1. Generative Models

Active inference rests upon a *generative model* of observable outcomes. This model is used to infer the most likely causes of outcomes in terms of expected states of the world. These states (and paths) are latent or *hidden* because they can only be inferred through observations. Some paths are controllable, in that they can be realised through action. Therefore, certain observations depend upon action, which requires the generative model to entertain expectations about outcomes under allowable policies. These expectations are optimised by minimising *variational free energy*. Crucially, the prior probability of a policy depends upon its *expected free energy*. Having evaluated the expected free energy of each policy, the most likely action is selected, and the implicit cycle of action and perception continues [1].

Figure 2 provides an overview of a generalised POMDP. Outcomes at any time depend upon hidden *states*, while transitions among hidden states depend upon *paths*. Note that paths are random variables, that may or may not depend upon action. The resulting POMDP is specified by a set of tensors. The first set **A**, maps from hidden states to outcome modalities; for example, exteroceptive (e.g., visual) or proprioceptive (e.g., eye position) *modalities*. These parameters encode the likelihood of an outcome given hidden states. The second set **B** encodes transitions among the hidden states of a *factor*, under a particular path. Factors correspond to different kinds of states, e.g., the location versus the class of an object. In the context of the worked example, one factor might represent the ball’s position, while another represents the paddle’s position. The remaining tensors encode prior beliefs about paths **C**, and initial conditions **D** and **E** referred to as hidden causes. The tensors are generally parameterised with independent Dirichlet distributions, whose sufficient statistics are Dirichlet counts, which count the number of times a particular combination of states and outcomes has been inferred. We will focus on learning the likelihood model, encoded by Dirichlet counts, a. When equipped with hierarchical depth, the model contains distinct tensors **A**, **B**, **C**, **D**, **E** at each level; symbols are reused at each level but do not denote the same tensors across levels. We summarise frequently used notation in Figure 2 and in Appendix A.

**Figure 2 entropy-27-00992-f002:**
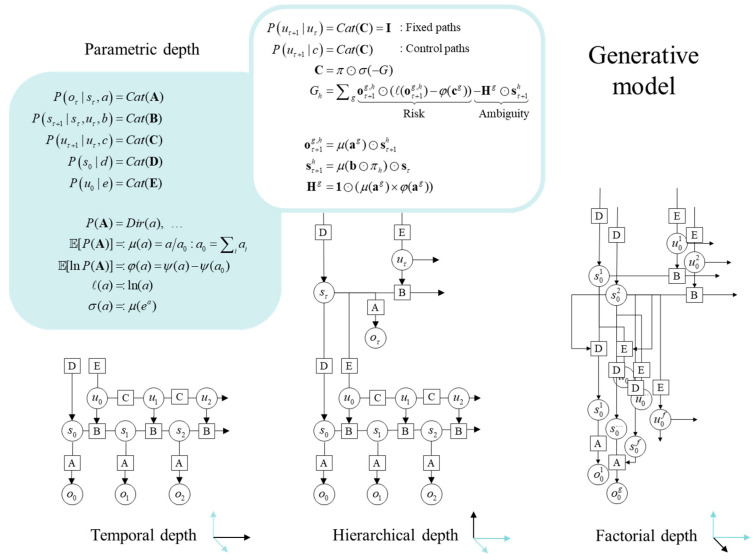
Generative models. A generative model specifies the joint probability of observable consequences and their hidden causes. Usually, the model is expressed in terms of a likelihood (the probability of consequences given their causes) and priors (over causes). When a prior depends upon a random variable it is called an empirical prior. Here, the likelihood is specified by a tensor A, encoding the probability of an outcome under every combination of states (s). Priors over transitions among hidden states, B depend upon paths (u), whose transition probabilities are encoded in C. Certain (control) paths are more probable a priori if they minimise their expected free energy (G), expressed in terms of risk and ambiguity (white panel). If the path is not controllable, it remains fixed over the epoch in question, where D and E specify the prior over initial states and paths, respectively. The left panel provides the functional form of the generative model in terms of categorical (Cat) distributions that are parameterised with Dirichlet (Dir) distributions, equipping the model with parametric depth. The lower equalities list the various operators required for variational message passing in Figure 3. These functions operate on each column of their tensor arguments. The graph on the lower left depicts the generative model as a probabilistic graphical model that foregrounds the implicit temporal depth implied by priors over state transitions and paths. This example shows dependencies for fixed paths. When equipped with hierarchical depth the POMDP acquires a separation of temporal scales. This follows because higher states generate a sequence of lower states. This means higher levels unfold more slowly than lower levels, furnishing empirical priors over hidden causes that contextualise the dynamics of their children. At each hierarchical level, hidden states and accompanying paths are factored to endow the model with factorial depth. In other words, the model ‘carves nature at its joints’ into factors that interact to generate outcomes (or the hidden causes of lower levels). Subscripts pertain to time, while superscripts denote distinct factors (f), outcome modalities (g) and combinations of paths over factors (h). Tensors and matrices are denoted by uppercase bold, while posterior expectations are in lowercase bold. We use the same symbols **A**, **B**, **C**, **D**, **E** at each hierarchical level; however, these are distinct at each level and dimensioned appropriately. The matrix *π* encodes the probability over paths, under each policy (for notational simplicity, we have assumed a single control path). The ⊙ notation implies a generalised inner (i.e., dot) product or tensor contraction, while × denotes the Hadamard (element by element) product. ψ(⋅) is the digamma function, applied to the columns of a tensor.

**Figure 3 entropy-27-00992-f003:**
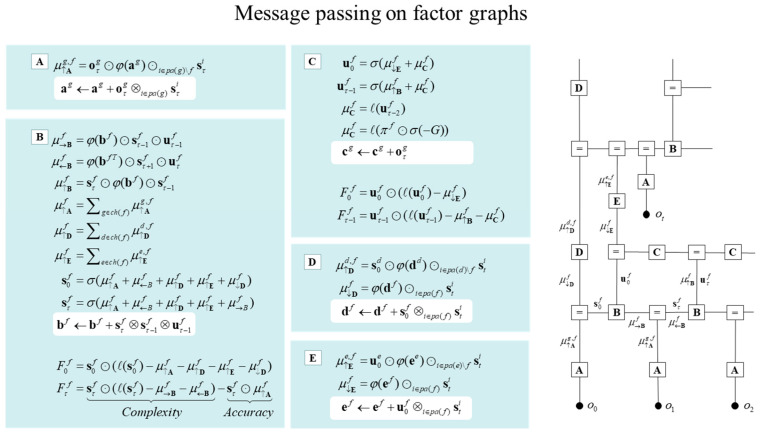
Belief updating and variational message passing: the right panel presents the generative model as a factor graph, where the nodes (square boxes) correspond to the factors of the generative model (labelled with the associated tensors). The edges connect factors that share dependencies on random variables. The leaves of (filled circles) correspond to known variables, such as observations (o). This representation is useful because it scaffolds the message passing—over the edges of the factor graph—that underwrite inference and planning. The functional forms of these messages are shown in the left-hand panels. For example, the expected path—in the first equality of panel C—is a SoftMax function of two messages. The first is a descending message $\mu_{↓E}^{f}$ from E that inherits from expectations about hidden states at the level above. The second is the log-likelihood of the path based upon expected free energy, G. This message depends upon Dirichlet counts scoring preferred outcomes—i.e., prior constraints on modality g—encoded in $c^{g}$: see Figure 2 and Equation (2). The two expressions for $\mu_{C}^{f}$ correspond to fixed and control paths, respectively. The updates in the lighter panels correspond to learning, i.e., updating Bayesian beliefs about parameters. Similar functional forms for the remaining messages can be derived by direct calculation. The ⊙ notation implies a generalised inner product or tensor contraction, while ⊗ denotes an outer product. ch(·) and pa(·) return the children and parents of latent variables.

The generative model in Figure 2 means that outcomes are generated by selecting a policy from a SoftMax function of expected free energy. Sequences of hidden states are then generated using the probability transitions specified by the selected combination of paths (i.e., policy). Finally, hidden states generate outcomes in one or more modalities. Model inversion updates the sufficient statistics—i.e., expectations (s,u,a)—of posterior beliefs $Q(s,u,a)=Q_{s}(s)Q_{u}(u)Q_{a}(a)$ that are factorised over hidden states, paths and parameters. This mean field factorisation effectively partitions belief updating into inference, planning and learning.

### 2.2. Variational Free Energy and Inference

In variational Bayesian inference (also known as approximate Bayesian inference), model inversion entails the minimisation of variational free energy *F* with respect to the sufficient statistics of approximate posteriors *Q*. For clarity, we will deal with a single factor, such that the policy (i.e., combination of paths) becomes the path, π=u. Omitting dependencies on previous states, we have for model m:(1)$$\begin{array}{l} Q(s_{\tau },u_{\tau },a)=\arg\min_{\;Q}F\\ \;\;\;\;\;\;\;\;\;\;\;\;\;\;\;\;\;F=E_{Q}[\ln\underset{\mathrm{Posterior}}{\underset{︸}{Q(s_{\tau },u_{\tau },a)}}-\ln\underset{\mathrm{Likelihood}}{\underset{︸}{P(o_{\tau }|s_{\tau },u_{\tau },a)}}-\ln\underset{\mathrm{Prior}}{\underset{︸}{P(s_{\tau },u_{\tau },a)}}]\\ \;\;\;\;\;\;\;\;\;\;\;\;\;\;\;\;\;\;\;=\underset{\mathrm{Divergence}}{\underset{︸}{D_{KL}[Q(s_{\tau },u_{\tau },a)||P(s_{\tau },u_{\tau },a|o_{\tau })]}}-\ln\underset{\mathrm{Evidence}}{\underset{︸}{P(o_{\tau })}}\\ \;\;\;\;\;\;\;\;\;\;\;\;\;\;\;\;\;\;\;=\underset{\mathrm{Complexity}}{\underset{︸}{D_{KL}[Q(s_{\tau },u_{\tau },a)||P(s_{\tau },u_{\tau },a)]}}-\underset{\mathrm{Accuracy}}{\underset{︸}{E_{Q}[\ln P(o_{\tau }|s_{\tau },u_{\tau },a)]}} \end{array}$$

Because the (KL) divergences cannot be less than zero, the penultimate equality means that free energy is zero when the (approximate) posterior is the true posterior. At this point, the free energy becomes the negative log evidence for the generative model [53]. This means minimising free energy is equivalent to maximising model evidence. Planning emerges under active inference by placing priors over (controllable) paths to minimise expected free energy *G:*(2)$$\begin{array}{l} G(u)=E_{Q_{u}}[\ln Q(s_{\tau +1},a|u)-\ln Q(s_{\tau +1},a|o_{\tau +1},u)-\ln P(o_{\tau +1}|c)]\\ \;\;\;\;\;\;\;\;\;=-\underset{\mathrm{Expected}\;\mathrm{information}\;\mathrm{gain}\;(\mathrm{learning})}{\underset{︸}{E_{Q_{u}}[\ln Q(a|s_{\tau +1},o_{\tau +1},u)-\ln Q(a|s_{\tau +1},u)]}}-\\ \;\;\;\;\;\;\;\;\;\;\;\underset{\mathrm{Expected}\;\mathrm{information}\;\mathrm{gain}\;(\mathrm{inference})}{\underset{︸}{E_{Q_{u}}[\ln Q(s_{\tau +1}|o_{\tau +1},u)-\ln Q(s_{\tau +1}|u)]}}\underset{\mathrm{Expected}\;\mathrm{cost}}{\underset{︸}{-E_{Q_{u}}[\ln P(o_{\tau +1}|c)]}}\\ \;\;\;\;\;\;\;\;\;=-\underset{\mathrm{Novelty}}{\underset{︸}{E_{Q_{u}}[D_{KL}[Q(a|s_{\tau +1},o_{\tau +1},u)||Q(a|s_{\tau +1},u)]]}}+\\ \;\;\;\;\;\;\;\;\;\;\;\underset{\mathrm{Risk}}{\underset{︸}{D_{KL}[Q(o_{\tau +1}|u)||P(o_{\tau +1}|c)]}}\underset{\mathrm{Ambiguity}}{\underset{︸}{-E_{Q_{u}}[\ln Q(o_{\tau +1}|s_{\tau +1},u)]}} \end{array}$$

Here, $Q_{u}=Q(o_{\tau +1},s_{\tau +1},a|u)=P(o_{\tau +1},s_{\tau +1},a|u,o_{0},\ldots ,o_{\tau })=P(o_{\tau +1}|s_{\tau +1},a)Q(s_{\tau +1},a|u)$ is the posterior predictive distribution over parameters, hidden states and outcomes at the next time step, under a particular path. Note that the expectation is over observations in the future, hence, expected free energy. This means that preferred outcomes—that subtend expected cost and risk—are prior beliefs, which constrain the implicit planning as inference: c.f., [59,60,61].

One can also express the prior over the parameters in terms of an expected free energy where, marginalising over paths:(3)$$\begin{array}{l} P(a)=\sigma (-G)\\ G(a)=E_{Q_{a}}[\ln P(s|a)-\ln P(s|o,a)-\ln P(o|c)]\\ \;\;\;\;\;\;\;\;=-\underset{\mathrm{Expected}\;\mathrm{information}\;\mathrm{gain}}{\underset{︸}{E_{Q_{a}}[\ln P(s|o,a)-\ln P(s|a)]}}\underset{\mathrm{Expected}\;\mathrm{cost}}{\underset{︸}{-E_{Q_{a}}[\ln P(o|c)]}}\\ \;\;\;\;\;\;\;\;=-\underset{\mathrm{Mutual}\;\mathrm{information}}{\underset{︸}{E_{Q_{a}}[D_{KL}[P(o,s|a)||P(o|a)P(s|a)]}}\underset{\mathrm{Expected}\;\mathrm{cost}}{\underset{︸}{-E_{Q_{a}}[\ln P(o|c)]}} \end{array}$$
where $Q_{a}=P(o|s,a)P(s|a)=P(o,s|a)$ is the joint distribution over outcomes and hidden states, encoded by Dirichlet parameters. Note that the Dirichlet parameters encode the mutual information, because they encode the joint distribution over outcomes and hidden states. When normalising each column of the *a* tensor, we recover the likelihood distribution (as in Figure 2). However, we could normalise over all entries of the tensor to recover a joint distribution, as in Equation (6).

Expected free energy can be regarded as a universal objective function that augments mutual information with expected costs or constraints. Constraints—parameterised by *c*—reflect the fact that we are dealing with open systems with characteristic outcomes, *o*. This can be read as a constrained principle of maximum mutual information or minimum redundancy [62,63,64,65]. In machine learning, this kind of objective function underwrites disentanglement [66,67], and generally leads to sparse representations [65,68,69,70].

### 2.3. Active Inference

In variational inference and learning, sufficient statistics—encoding posterior expectations—are updated to minimise variational free energy. Figure 3 illustrates these updates in the form of variational message passing. For example, expectations about hidden states are a SoftMax function σ of messages that are linear combinations of other expectations and observations.(4)$$\begin{array}{l} \;\;s_{\tau }^{f}=\sigma (\mu_{↑A}^{f}+\mu_{\to B}^{f}+\mu_{\leftarrow B}^{f}+\ldots )\\ \mu_{↑A}^{f}=\sum_{g\in ch(f)}\mu_{↑A}^{g,f}\\ \mu_{↑A}^{g,f}=o_{\tau }^{g}⊙\varphi (a^{g}){⊙}_{i\in pa(g)\backslash f}s_{\tau }^{i} \end{array}$$

Here, $\mu_{↑A},\mu_{\to B},\mu_{\leftarrow B}$ denote ascending, forward and backward messages from likelihood and transition tensors, respectively, which may be indexed by latent state factor *f* or modality *g*. See Appendix A for a glossary of notation. Here, the ascending messages from the likelihood factor are a linear mixture of expected states and observations, weighted by (digamma) functions of the Dirichlet counts that correspond to the expected likelihood, i.e., normalised (φ) Dirichlet counts (c.f., connection weights). (The ⊙ notation implies a sum product operator, i.e., the dot or inner product that sums over one dimension of an array or tensor. In this paper, these operators are applied to a vector a and a tensor A where a⊙A implies the sum of products is taken over the leading dimension, while A⊙a implies the sum is taken over a trailing dimension. For example, 1⊙A is the sum over rows and $A⊙1=A_{0}$ is the sum over columns, when 1 is a vector of ones and A is a matrix. If A is a matrix, then $a⊙A=a^{T}\cdot A$. Finally,$A_{•,i}$ and $A_{i,•}$ refer to the i-th column and row of A, respectively. This notation replaces the Einstein summation notation to avoid visual clutter). The expressions in Figure 3 are effectively the fixed points (i.e., minima) of variational free energy. This means that message passing corresponds to a gradient free, fixed-point iteration scheme [53,71,72] (Figure 3 and Equation (4) provide both the forward and backward messages in time. In practice—and in the examples below—it is sometimes simpler to omit backward messages; in which case the forward messages can be replaced with exact updates).

### 2.4. Active Learning

In the setting of discrete models, learning corresponds to updating model parameters by accumulating Dirichlet counts based upon posterior expectations. (Technically, this corresponds to a coordinate ascent variational inference (CAVI) update). In our worked example, if the agent infers the ball is at a certain location (a hidden state *s*) while observing a corresponding configuration of pixels (an outcome *o*), the corresponding entry in the Dirichlet count tensor **a** for the likelihood mapping that links this state to this outcome is incremented. For example, for the likelihood tensors we have:(5)$$a^{g}\leftarrow a^{g}+o_{\tau }^{g}{⊗}_{i\in pa(g)}s_{\tau }^{i}$$

Active learning has a specific meaning here. It implies that the updating of Dirichlet counts depends on expected free energy; namely, the mutual information encoded by the tensors: see Equation (1). This means that an update is selected in relation to expected information gain. Consider two actions: to update or not to update. From Equation (5), we have (dropping the modality superscript for clarity):(6)$$\begin{array}{l} \Delta a=o_{\tau }{⊗}_{i\in pa}s_{\tau }^{i}\\ E[P(o,s|a)]=\bar{a}=:\frac{a}{\Sigma (a)}\;\\ E_{Q}[a|u=u_{o}]=a|u_{o}=a\\ E_{Q}[a|u=u_{1}]=a|u_{1}=a+\Delta a \end{array}$$

Here,  Σ(a) is the sum of all tensor elements. The prior probability of committing to an update is given by the expected free energy of the respective Dirichlet parameters, which scores the expected information gain (i.e., mutual information) and cost. (For simplicity, we assume tensors have been formatted as matrices, by vectorising the second and trailing dimensions):(7)$$\begin{array}{l} P(u)=\sigma (-\alpha \cdot G(a|u))\\ G(a)=-\underset{\mathrm{Mutual}\;\mathrm{information}\;(\mathrm{relative}\;\mathrm{entropy})}{\underset{︸}{E_{Q_{a}}[D_{KL}[P(o,s|a)||P(o|a)P(s|a)]}}\underset{\mathrm{Expected}\;\mathrm{cost}\;(\mathrm{cross}\;\mathrm{entropy})}{\underset{︸}{-E_{Q_{a}}[\ln P(o|c)]}}=C(a)-I(a)\\ I(a)=\underset{\mathrm{Marginal}\;\mathrm{entropies}}{\underset{︸}{\left(1⊙\bar{a})⊙l(1⊙\bar{a})-(\bar{a}⊙1)⊙l(\bar{a}⊙1\right)}}-\underset{\mathrm{Joint}\;\mathrm{entropy}}{\underset{︸}{1⊙(\bar{a}\times l(\bar{a}))⊙1}}\\ C(a)=\underset{\mathrm{Cross}\;\mathrm{entropy}}{\underset{︸}{-\varphi (c)⊙(\bar{a}⊙1)}} \end{array}$$
This prior over the updates furnishes a Bayesian model average of the likelihood parameters, effectively marginalising over update policies:(8)$$\begin{array}{l} E_{Q}[a_{\tau +1}^{g}]=P(u_{0})\cdot E_{Q}[a_{\tau }^{g}|u=u_{o}]+P(u_{1})E_{Q}[a_{\tau }^{g}|u=u_{i}]\\ \;\;\;\;\;\;\;\;\;\;\;\;\;\;\;\;\Rightarrow \\ \;\;\;\;\;\;\;\;a_{\tau +1}^{g}=P(u_{0})\cdot a_{\tau }^{g}+P(u_{1})\cdot (a_{\tau }^{g}+\Delta a_{\tau }^{g})\\ \;\;\;\;\;\;\;\;\;\;\;\;\;\;\;\;=a_{\tau }^{g}+P(u_{1})\cdot \Delta a_{\tau }^{g} \end{array}$$

In Equation (7), α plays the role of a hyperprior that determines the sensitivity to expected free energy. When this precision parameter is large, the Bayesian model average becomes Bayesian model selection, i.e., either the update is selected, or it is not. Active learning of this sort rests on treating an update as an action that is licenced if expected free energy decreases. A complementary perspective—on this selective updating—is that it instantiates a kind of Maxwell’s Demon; selecting just those updates that maximise (constrained) mutual information. Exactly the same idea can be applied to model selection, leading to active (model) selection.

### 2.5. Active Selection

In contrast to learning—that optimises *posteriors* over parameters—Bayesian model selection or structure learning [38,73,74] can be framed as optimising the *priors* over model parameters. On this view, model selection can be implemented efficiently using Bayesian model reduction. Bayesian model reduction is a generalisation of ubiquitous procedures in statistics [75]. By applying Bayes rules to parent and reduced models it is straightforward to show that the change in variational free energy can be expressed in terms of posterior Dirichlet counts **a**, prior counts *a* and the prior counts that define a reduced model *a*’. Using Β to denote the beta function, we have [76]:(9)$$\begin{array}{l} \Delta F=\ln P(o|a)-\ln P(o|a^{\prime })\\ \;\;\;\;\;=\ln Β(a)+\ln Β(a^{\prime })-\ln Β(a)-\ln Β(a+a^{\prime }-a)\\ \;\;a^{\prime }=a+a^{\prime }-a \end{array}$$

Here, $a^{\prime }$ corresponds to the posterior that one would have obtained under the reduced priors. The alternative to Bayesian model reduction is the bottom-up growth of models to accommodate new observations. If one considers the selection of one (parent) model over another (augmented) model as an action, then the difference in expected free energy furnishes a log prior over models that can be combined with the (variational free energy bound on) log marginal likelihoods to score their posterior probability. This can be expressed in terms of a log Bayes factor (i.e., odds ratio) comparing the likelihood of two models, given some observations, o:(10)$$\begin{array}{l} \ln\frac{P(m|o)}{P(m^{\prime }|o)}=\ln\frac{P(o|m)}{P(o|m^{\prime })}+\ln\frac{P(m)}{P(m^{\prime })}=\Delta F+\Delta G\\ \;\;\;\;\;\;\;\;\;\;\;\;\;\;\Delta F=\ln P(o|m)-\ln P(o|m^{\prime })\\ \;\;\;\;\;\;\;\;\;\;\;\;\;\;\Delta G=\ln P(m)-\ln P(m^{\prime })=G(a|m)-G(a^{\prime }|m^{\prime }) \end{array}$$

Here, a and $a^{\prime }$ denote the posterior expectations of parameters under a parent m, and augmented model $m^{\prime }$, respectively. One can now retain or reject the parent model, depending upon whether the log odds ratio is greater than or less than zero, respectively. Active model selection therefore finds structures with precise likelihood mappings and transition priors. In effect, when assimilating new (e.g., training) data one can simply equip the model with a new hidden state to explain every unique observation [39]. This affords a fast kind of structure learning. We now turn to how fast structure learning can be used to discover pullback attractors that evince goal-directed behaviour.

## 3. Attractor Learning

This section rehearses the model selection and learning procedures that are illustrated in the next section. In brief, de novo structure learning—as described here—has three phases. First, the structure of the model is learned from observations generated under random actions (in our worked example, by moving the paddle left or right at random). This must be the starting point, because there is no generative model for policy selection. One could provide curated training data—in the spirit of supervised structure learning—that enables the agent to discover factorial structure and compositionality [39]. However, here, we assume there is no supervision and the agent has to learn the cause-effect structure of its world from scratch. In the parlance of the free energy principle, we assume an agent starts with its Markov blanket—i.e., sensory and active states—but with nothing on the inside, to mediate between the two.

The initial phase of learning simply identifies the structure of the model in terms of edges that encode dependencies of outcomes (i.e., observations and hidden causes) on hidden states. In the second phase, the Dirichlet parameters of the likelihood and transition tensors are accumulated under this structure if, and only if, epochs of observations include one or more rewarded outcomes (e.g., an outcome that includes hitting a central target). This can be viewed as a simple form of active learning; in the sense the absence of reward precludes parameter learning, via the expected cost in Equation (7). Crucially, after each epoch the model is reduced to retain states that lie within the basins of attraction of (goal) states associated with rewards. This furnishes a model of closed orbits through successive goal states. The resulting generative model can then be used in a third phase for continual (structural and parametric) learning, using inductive inference [48] for sequential policy optimisation. Continual learning allows the agent to finesse its model in light of previously un-encountered outcomes and become progressively confident in its policy selection. We now consider these phases in detail.

### 3.1. Phase 1: Structure Learning

The first thing we need to identify is the structural form of the generative model. This form can be cast as a probabilistic graphical model defined in terms of nodes (i.e., latent states) and edges (i.e., conditional dependencies). The latent causes of observations are assumed to be discrete. However, in dealing with pullback attractors, we need to generalise the notion of a state to include paths or transitions as in Figure 2. In continuous settings, this induces states in generalised coordinates of motion [77,78]. The equivalent discrete representation is afforded by generalised Markov decision processes. In these generalised schemes, the state at a higher level generates the initial state and transitions at a lower level (generalised states could be thought of as deep states). In the setting of renormalising generative models—where the same operations are repeated at successive hierarchical levels—generalised states encode increasingly higher orders of transitions at lower levels, i.e., paths, paths of paths, and so on.

We will use renormalising generative models to instantiate a scale-free implementation of the Bayesian belief updating described in Figure 3. In these models, hidden states at each hierarchical level generate the hidden causes of the level below. These hidden causes are the initial states and paths of subordinate levels, with observations at the lower level: see Figure 4. Hidden causes or outcomes $s_{0}^{\left(i-1\right)},u_{0}^{\left(i-1\right)}\in o^{\left(i\right)}$ at level i are partitioned such that each subset (i.e., group) comprising that partition is generated by a (generalised) state at the level above. This has the important consequence that there are no latent causes with more than one parent. In turn, this means that all latent states are conditionally independent, given the level below, thereby finessing the size of requisite likelihood matrices $D^{\left(i-1\right)},E^{\left(i-1\right)}\in A^{\left(i\right)}$, which map between levels. To realise this parametric efficiency, one has to specify which higher-level latent state serves as the (single) parent of each lower-level variable: namely, hidden states $id.D^{f},id.E^{f}\in pa(B^{f})$ and causes $id.A^{g}\in pa(A^{g})$. These parents (i.e., graphical edges) define the dependency structure of the model and are discovered in the first phase of structure learning. The dependence of lower-level groups on a single parent has the added benefit that it limits the number of cycles in the graph, which supports exact (no cycles) or close-to-exact (few cycles) marginal posteriors using fixed point belief-propagation [79]. This also ensures a compression at each level such that we move from microscale variables, via mesoscale, to more macroscopic variables. Each of these levels, during inference and learning, constrains the dynamics of adjacent levels bidirectionally—closely echoing ideas about the collective emergence of metastability in brain function [80].

Practically, this involves partitioning a set of hidden causes at any hierarchical level—or outcomes at the lowest level—such that each group maps to a hidden state at the level above in a way that maximises mutual information, i.e., minimises expected free energy or information loss. Effectively, this simply entails identifying all the unique combinations of outcomes, within a group that one expects to encounter. By populating the likelihood mappings with Dirichlet counts of the number of unique occurrences—following Equation (6)—one effectively implements a lossless compression. One can identify unique combinations in a variety of ways. For example, one can identify the unique columns of a dissimilarity matrix Δ∈ℝT×T whose elements are the information distance among T outcomes, where each outcome is a group of G marginals o, over hidden causes or observations:(11)     O=o11…oT1⋮⋮o1G…oTG︸Grouped outcomesΔ(O)=2⋅2⋅G⋅(I−η(O)⊙η(O)))︸Information distance     k=unique(Δ(O)<1)    Rij=1:j=ki0:j≠ki     ag=[o1g,…,oTg]⋅R

Here, the elements of k∈ℤ1×T index unique instances and R∈[0,1]T×K is a (logical) reduction matrix that combines T outcomes from one group into K unique instances. Unique instances are those that are within one natural unit (2.72 bits) of each other. Information distances among outcomes can be computed efficiently following (Euclidean) normalisation η(O), which sets the sum of squares of each column to one. Effectively, the information distance scores the distance between categorical distributions on a hypersphere: see the appendix of [81] for details. However, when dealing with precise marginals (and observations), an even simpler approach is to find unique instances of nonzero probabilities in the columns of concatenated outcomes:(12)k=unique(O<1)

Having compressed a sequence of outcomes into a likelihood matrix, the corresponding compression of dynamics simply involves accumulating all unique transitions in a transition tensor, where each unique transition from any state induces an extra path (i.e., an extra ‘slice’ or ‘slot’ of the transition tensor). This compresses a sequence of outcomes—at any given level—into unique states and paths, where the number of paths corresponds to the size of the third dimension of the transition tensors. Using this procedure, one can compress outcomes into sequences of unique states that are decimated to provide hidden causes $s_{0}^{\left(i-1\right)},u_{0}^{\left(i-1\right)}\in o^{\left(i\right)}$ (i.e., initial states and paths) for compression at the next level. Here, we take every second hidden cause. This process is repeated until there is one group at the final level. (Not all observations are equipped with parents. For example, if certain modalities have no dynamics, then there is no shared (mutual) information with their parents (whose transition tensor reduces to a single element of one). A common example of this would be background pixels in an image that never change. Removing the parents of uninformative outcomes ensures a high mutual information via lossless compression). Please see [58] for a detailed description of this renormalisation procedure.

Usually, the partitioning of outcomes into groups calls upon some knowledge about the statistical dependencies among outcomes. Previously, we illustrated the use of spin-block partitioning based upon the location of pixels in an image [58,82]. This presupposes that neighbouring pixels are conditionally dependent via local interactions or contiguity constraints. In this work, we eschew such assumptions and partition outcomes based upon their mutual information. The mutual information $\Omega_{ij}$ between outcomes i and j can be evaluated from a sequence of observations as follows, where I is defined as follows:(13)$$\Omega_{ij}=I\left(\sum_{t}o_{t}^{i}⊗o_{t}^{j}\right)$$

The ensuing matrix is, by construction, positive definite with non-negative elements. From the Perron–Frobenius theorem, the principal eigenvector $\varepsilon =\lambda^{-1}\Omega \varepsilon$ has strictly positive values and can be regarded as the (unnormalised) probability that each element belongs to the principal group. This is the basis of spectral clustering in graph theory [83]. Here, we use spectral clustering to remove the principal group and then repeat the procedure, until all outcomes have been assigned to a group and, implicitly, their parents. One can place an upper bound on the size of each group that specifies the degree of coarse graining. We use an upper bound of 32 in this paper. This ensures that all requisite likelihood mappings in the generative model are two-dimensional tensors (i.e., matrices) with leading dimensions of 32 or less.

In many instances, observations can be partitioned into distinct classes or streams. For example, in neurobiology, observation modalities come in three flavours: exteroceptive, interoceptive and proprioceptive. In artificial agents, this partitioning could be read as visual input comprising pixels, reward and punishment streams—used to signal preferred and costly outcomes—and, finally, telemetry or inertial measurement units (IMU), used to report the states of an actuator. The existence of distinct streams can be accommodated within renormalising models by keeping each stream separate, until the final level, where all hidden causes are assigned a single parent. This affords a compressed representation of generalised states that can generate outcomes in all modality streams. This multisensory integration is a common feature in computational neuroscience [84,85,86] and enables the causes of one sensory stream to be recognised and thereby furnish predictions of another. This is particularly useful if the agent is equipped with the ability to realise predictions in the proprioceptive stream. This enables (motor) control under active inference, in the spirit of the equilibrium point hypothesis and model predictive control [87,88]. We will see examples of this later. Figure 4 illustrates this multisensory or multimodal architecture in the simple case of two streams, where the second stream has a single outcome modality.

To summarise, structure learning takes a sequence of outcomes, each encoded as one-hot categorical probability distributions, and compresses them into likelihood and transition tensors whose dependencies are specified in terms of edges or parents. These parents specify the structural form of the generative model. The next step is to learn the functional form by populating the tensors with Dirichlet parameters.

### 3.2. Phase 2: Attractor Learning

Having established the structure of the generative model in terms of edges or dependencies, observations can now be assimilated to learn the likelihood and transition tensors, in accord with Equation (12). However, active learning mandates that this assimilation is selective, in the sense it minimises expected cost according to Equation (7). One can implement this selective updating by exposing the model to sequences—generated under random action—that contain one or more rewarded outcomes. However, this incurs the possibility of learning transitions to states that generate costly outcomes, which need to be removed after each training epoch.

One could see this as analogous to a kind of natural selection in which random mutation (sampling random actions) is interleaved with selection (removal of costly examples). Heuristically, this sort of dynamic can be articulated in terms of ‘Darwinian’ dynamics’ [89] of a sort closely related to the Helmholtz decomposition that supports conservative changes (e.g., mutations that do not change overall fitness) in addition to the dissipative dynamics that eliminate phenotypes with lower evolutionary fitness.

One can generalise this (Bayesian) model reduction to removing transitions that do not lead to (or from) a goal state. A goal state is a hidden state that generates a sequence of observations with one or more rewards; in our game, this would be a generalised state that includes the ball hitting a central target. Generalised states that do not lead to—or follow from—a goal state can be identified easily by asking whether they lie in the basins of attraction of a goal state. These basins or orbits are identified in a straightforward way by propagating goal states backwards—and forwards—in time using the highest-level transition tensor (collapsed over the path dimension). (This reachability test does not assume deterministic dynamics. A state is deemed reachable if any path with non-zero probability exists, so stochastic and deterministic environments are treated identically. The analysis therefore identifies pullback basins of attraction in the sense of random dynamical systems theory; the only limitation is that very low-probability transitions may be missed if they are never observed during training).(14)$$\begin{array}{l} B_{ij}=\neg \chi_{i}\wedge \left(\sum_{k}b_{ijk}\right)>1\\ \;l_{t}^{-}=\gamma ⊙B^{t}>1\\ \;l_{t}^{+}=B^{t}\cdot \gamma >1\\ \;\;k=\underset{\mathrm{Cost}-\mathrm{free}\;\mathrm{and}\;\mathrm{transitive}}{\underset{︸}{\neg \chi \wedge \overset{\mathrm{Forward}\;\mathrm{or}\;\mathrm{backward}\;\mathrm{basins}\;\mathrm{of}\;\mathrm{attraction}}{\overset{︷}{\left[\left(\sum_{t}l_{t}^{-}\right)>1\vee \left(\sum_{t}l_{t}^{+}\right)>1\right]}}}}\\ R_{ij}=\left\{\begin{array}{cc} 1 & :j=k_{i}\\ 0 & :j\neq k_{i} \end{array}\right. \end{array}$$

Here, $\chi \in {\left[0,1\right]}^{I\times 1}$ and $\gamma \in {\left[0,1\right]}^{I\times 1}$ are logical vectors encoding costly and goal states, respectively. The first equality creates a transition matrix of possible state transitions but precluding transitions to costly states. The last equality creates a (reduction) matrix R of logical values that is used to reduce the highest-level transition tensor and accompanying likelihood tensors of Dirichlet counts:(15)$$\begin{array}{c} a_{•,•}^{g}\leftarrow a_{•,•}^{g}\cdot R\\ b_{•,•,k}^{f}\leftarrow R⊙b_{•,•,k}^{f}\cdot R \end{array}$$

The forward and backward basins of attractions are depicted in Figure 5 in terms of ancestors (i.e., predecessors) and descendants (i.e., successors) of goal states. The values of t in Equation (14) determine the size of the basins (i.e., length of forward and backward orbits) and may vary from application to application. In this work, we use 32 transitions, i.e., a newly discovered transition is retained if there exists a path to or from a goal state within 32 transitions or less. This affords a degree of control over the size of the transition tensor at the highest level. The third dimension of this tensor can never exceed the number of generalised states. This follows because the number of paths corresponds to the maximum number of states any state can transition to, which cannot be greater than the number of states.

This form of model reduction retains generalised states that lie within the basins of attraction of goal states; thereby affording the opportunity to access these goal states from every generalised state the model entertains. By repeating successive iterations of growing and reducing the model in this way, generalised states within basins of attraction are selectively accumulated. At some point the forward orbits of one goal state will fall in the basin of attraction of another, thereby furnishing a path from one goal state to the next. With a sufficient number of accumulated transitions, a subset of goal states will be mutually accessible. Technically speaking, this means a subset of discovered states have transitions that are topologically transitive. In other words, the generative model has discovered transitions among generalised states that support a pullback attractor, i.e., a dense and closed orbit among goal states. See Figure 5 for a schematic illustration of this process.

Note that this kind of learning does not require exposure to how goals are reached. The model just needs to see rewarded or punished outcomes and retain certain state transitions. These transitions may or may not be used in the order they were discovered. In other words, the model accumulates possibilities that are sufficient to support goal-directed behaviour, as opposed to simply accumulating instances of rewarded behaviour. It is these possibilities that offer affordances for action in the final phase of (continual) learning.

This kind of attractor learning rests on selectively accumulating transitions that are sufficient to generate goal directed behaviour that can be sustained indefinitely. In other words, the model has selectively discovered a small subset of transitions relative to all possible transitions. This is important, because the model can only learn goal-directed transitions. This can be contrasted with a non-selective learning of all allowable transitions and then selecting those transitions or policies to produce goal-directed behaviour. For example, in the numerical example of the next section, there were 258,552 allowable transitions—of 14.7 billion possible transitions—that characterise the physics of the game, while only 0.4% (i.e., ~1000 transitions) are required for expert play. Heuristically, attractor learning would be like learning to ride a bike, without learning all the ways to fall over. This analogy brings us to the final part of attractor learning.

Having learned generalised transitions that are sufficient to support a pullback attractor, one can now further compress the model by eliminating any generalised states that do not have definitive successors, i.e., generalised states from which there is no learned recovery (e.g., crashing your bike irrevocably). Intuitively, these ‘dead ends’ can be removed by recursively eliminating generalised states with no successors. After this pruning, all generalised states lie on the attractor or its insets, where the first generalised state of an inset has no ancestors. These ‘orphan’ states correspond to initial conditions that portend subsequent goal-directed behaviour. Having pruned the model, we now turn to the third phase in which the model is equipped with agency to generate its own outcomes for autodidactic (continual) learning.

### 3.3. Phase 3: Continual Learning

The final phase of learning simply repeats the iterations of accumulating novel transitions and subsequent model reduction but replacing outcomes under random actions with actions selected under active inference. In our worked example, this means that instead of moving the paddle randomly, the agent now uses its nascent model of the game to choose moves that it predicts will lead to rewards and then uses the consequences of those actions to further refine its model. This means we turn the generative model into an agent by equipping it with the belief that the generalised transitions at the highest (multimodal) level are controllable. By identifying goal states, the agent can now use inductive inference to find the path of least action—i.e., the shortest path—to a goal state. In this setting, a policy is just a sequence of switches among the paths encoded by the transition tensor at the highest level. The most likely policy identifies the next generalised state. In turn, this furnishes empirical priors over outcomes at the next time point. Crucially, these predictions include the states of the agent (or its actuators). This means that the agent can select the action that maximises the accuracy of outcomes under the predictive posterior. (This is the only way that action can minimise variational free energy, because action can only change observations: see the final equality in Equation (1)). In summary, by simply rendering generalised states at the highest level controllable, the agent will—a priori—believe it is pursuing paths of least action from goal to goal and realise the ensuing predictions by acting or moving. Usually—and in the numerical studies below—predictions are realised by selecting the action that maximises the accuracy of proprioceptive observations (e.g., the position of an agent or its manipulanda), under their predictive posterior: c.f., classical reflex arcs in neurobiology [90,91].

The resulting observations can now be used for (continual) parameter learning as above to reinforce the paths of least action. This phase of learning enables the agent to learn from mistakes or specialise its generative model to any stochastic or previously unseen transitions. In other words, it can assimilate novel transitions, provided they converge on the agent’s pullback attractor. As the agent learns that its path on the attractor manifold is the shortest path between successive goal states, it also becomes more confident in its predictive posterior beliefs and subsequent policy selection: see Equation (2).

To invoke active inference, it is necessary to reduce the precision of empirical priors, such that belief updating becomes sensitive to observations via their likelihood; namely, the messages $\mu_{↑A}^{f}$ in Equation (4). This is simple to implement by adding a small concentration parameter—e.g., 1/512—to the transition tensors at each and every level of the model. This ensures that precise beliefs about the next generalised state do not overwrite the sensory evidence at hand (that inherits from precise likelihood mappings).

Finally, after the agent becomes fluent and confident, one can apply a further model reduction procedure that preserves the mutual information between generalised states at the highest level and their consequences, namely outcomes in the interoceptive and proprioceptive streams (i.e., action and its consequences). Effectively, this merges generalised states that predict the same interoceptive and proprioceptive states, now and in the future: c.f., [92,93,94]. To merge generalised states in a way that preserves mutual information, one can use unique information distances to retain mixtures of generalised states that predict current and subsequent outcomes in streams reporting the consequences of action:(16)$$\begin{array}{l} \;\;\;\;\;O=\left[\begin{array}{c} O_{2}\\ \vdots \\ O_{S} \end{array}\right],\;O_{s}=\underset{\mathrm{Likelihood}\;\mathrm{mappings}}{\underset{︸}{\left[\begin{array}{l} \mu (d^{g\in s})\\ \mu (e^{g\in s})\\ \mu (d^{g\in s})\cdot \mu (b^{f})\\ \mu (e^{g\in s})\cdot \mu (b^{f}) \end{array}\right]}}\\ \;\;\;\;\;\;k=unique(\Delta (O)<1)\\ \;\;\;\;R_{ij}=\left\{\begin{array}{cc} 1 & :j=k_{i}\\ 0 & :j\neq k_{i} \end{array}\right.\\ a^{g\in s}\leftarrow a^{g\in s}\cdot R\\ \;\;\;b^{f}\leftarrow R⊙b^{f}\cdot R \end{array}$$

Here, $\mu (b^{f})$ is the normalised probability transition tensor at the highest level (for simplicity, we have assumed a single path) and $d^{g\in s},e^{g\in s}\in a^{g\in s}$ are the unnormalised likelihood mappings (i.e., Dirichlet matrices) to the initial states and paths of stream s. Note that the above model reduction discounts the first stream, under the assumption that the information required for inductive inference is contained in subsequent (i.e., interoceptive and proprioceptive) streams that report the consequences of action.

The implicit model reduction has the interesting consequence that generalised states now predict one or more combinations of outcomes. If one reads the tensor operations entailed by belief updating as logical operators, then this is the same as assigning (reduced) generalised states to one outcome OR another, as opposed to the uncompressed model in which each generalised state specifies a unique outcome at the lowest level. In other words, certain generalised states and paths are merged in a way that preserves the mutual information between the current state of play and the future; thereby enabling inductive inference. In the example below, we used inductive inference over 64 transitions into the future. Because inductive inference uses logical tensor operations, sequential policies can be selected comfortably at superhuman speeds.

## 4. A Worked Example

This section applies the procedures of the previous section to a bespoke arcade game. We created a small game that offers several challenges that characterise these games: namely, occlusions, interacting objects, random resets, moving objects to be avoided, the appearance and disappearance of targets, sparse rewards, dense costs, and so on. In brief, the game can be regarded as a combination of classic arcade games, in which a ball bounces around inside a bounded box and can be hit in certain ways to eliminate rows of targets at the top of a (12 × 9) pixel array. Each time a row of targets is hit, it disappears, revealing the row behind it, until all rows have been hit and the game is reset. This reset reinstates the targets and places the ball near the centre of the frame at a fixed height. The paddle that can be moved to the right or left along the lowest row of pixels. Crucially, the paddle is sticky and confers its momentum to the ball. In other words, if the paddle is moving to the right the ball will be returned with a rightward velocity. If the paddle is stationary, the ball is returned on a vertical trajectory, and so on.

Although targets are removed when any target in a row is hit by the ball, only the central targets elicit a reward. Losses are incurred when the ball is missed, at which point the position of the ball is reset at random to one of three locations, as described above. The agent has to play this game with the added complication that (three) bombs periodically appear a few (four) rows above the paddle and then descend. If the bomb lands on the paddle, a loss is incurred, unless the ball is at the same location to protect the paddle. This game features the usual challenges for reinforcement learning, e.g., the temporal credit assignment problem induced by obtaining sparse rewards many timesteps after the ball is returned. Furthermore, it features action at a distance, which confounds the use of spatial or metric information, e.g., an entire row of targets disappears, irrespective of whether a rewarding target is hit.

The requisite game engine was specified—using logical statements—to build the requisite transition tensors that generate the next state of play, and consequent states of pixels, given the previous state and one of three actions (move the paddle to the right, to the left or leave it where it is). The game has 111 observation channels (i.e., modalities; 12 × 9 = 108 for pixels, and 3 for rewards, punishment and paddle location).

With this game at hand, we now illustrate de novo learning and subsequent intentional behaviour.

### 4.1. Phase 1: Structure Learning

To discover the structure of the model, it is necessary to determine which modalities should be grouped under their latent states, and which hidden causes of those latent states should be grouped for subsequent renormalisation. In other words, the compositional structure of the generative model has to be discovered—expressed in terms of conditional dependencies. This requires exposure to a sufficiently long sequence of random play for spectral clustering at each hierarchical level.

The size of this problem depends on the number of latent states and paths necessary to specify the cause-effect structure of the game. Here, there were 69,984 possible states and 258,552 allowable transitions. Although this is a small fraction of all theoretically possible transitions, (0.0018%), it would require many exposures to see all allowable transitions. (The length of the training sequence is determined largely by the physics of the game. In particular, it is necessary to present a sufficiently long sequence to include rare outcomes. This is because if any outcome does not include at least one transition, its latent state will have no parents. Latent states with no parents (e.g., background pixels that never change) can still be generated and recognised; however, they cannot inform, or be informed, by deeper levels of the generative model). However, because we can upper bound the size of groups, one can estimate the joint distribution over groups of outcomes—here, 32 or less—with a relatively short sequence. In this example we used 10,000 frames of gameplay.

Structure learning automatically partitioned the 111 modalities (12 × 9 = 108 for pixels, and 3 for rewards, punishment and paddle location) into 19 groups, each with their own latent state. The hidden causes of these factors were then partitioned into 12 groups: 6 for the pixel stream and 6 for the three remaining streams. These 12 latent causes correspond to 6 pairs specifying the initial states and paths at the first level. At the next (second) hierarchical level, the 12 latent ‘outcome’ variables—namely the six initial-state/path pairs—were all assigned a common parent factor of generalised states: c.f., Figure 4. This produced a 2-level model, specified in terms of the parents of latent variables and observations. The tensors encoding these factors were emptied of Dirichlet counts, ag∈ℝ0×0, bf∈ℝ0×0×0, prior to populating them in the second phase of de novo learning.

### 4.2. Phase 2: Attractor Learning

Equipped with the causal architecture, specified in terms of edges or parents, one can now accumulate states and transitions to grow the model, under the constraint that paths through generalised state-space pass through goal states and avoid costly states. In this example, we exposed the model to 100-frame epochs or games that included a reward. The model was reduced after each game according to Equation (15), retaining generalised states in basins of attraction (with an upper bound of 32 transitions to or from a goal state). This process was repeated until the emergence of a pullback attractor encompassing at least 64 goal states. At this point, the model was pruned to remove absorbing states in preparation for continual learning.

Figure 6 reports an example of attractor learning in which a pullback attractor of the requisite size was discovered after ~170 games. The upper panels show the number of generalised states and paths assimilated as a function of the number of games; here, over 2000 states and 9 paths. There are 9 paths because the three (random) actions can be followed by three subsequent actions. Of the 2000+ generalised states about 350 are absorbing, i.e., the final states of a game, whose successors have never been observed. Similarly, there are about 250 orphan states; namely, initial states whose predecessors have never been witnessed. This means that the majority of generalised states are transient states, which include ~280 goal states. Of these goal states, about 110 are transient goal states, in the sense there is a path to a subsequent goal state. Of these transient goal states, 64 can be accessed from each other. These constitute goal states on a pullback attractor or closed orbit. The lower left panel illustrates the discovered transitions, among generalised states, in the form of an adjacency matrix. The lower right panel shows the equivalent adjacency matrix for goal states, indicating a path from each of the 280 goal states to the others (within 32 transitions). One can see that the majority of goal states have no successors. These goal states are removed during the pruning of absorbing states.

Figure 7 shows goal adjacency matrices before (upper left panel) and after (upper right panel) pruning. About 100 goal states survive pruning. These goal states are on the pullback attractor, as illustrated in the lower panels that show the implicit paths as directed graphs before (middle row) and after (lower row) transitive reduction. Transitive reduction returns a directed graph with the same vertices but with the minimum number of edges, producing the smallest directed graph in which reachability from one state to another is preserved. In other words, the reduction is a directed graph that has the same reachability relations as the directed graph, i.e., the same transitive closure.

In this example, there are two pairs of closed orbits, where one has a number of insets (corresponding to goals that cannot be reached from other goals): see lower right panel. It is interesting to note that the two orbits cannot be reached from each other. This follows from the renormalising structure of the generative model. In brief, because generalised states represent paths with initial and final states, there are (degenerate) paths that start at each time point along the path. Here, the generalised states cover two timesteps and there are two (degenerate) orbits. Figure 7 shows that prior to pruning, there are many goal states that cannot be reached by, or reach, other goal states. These are shown as isolated vertices in the left panels. These goal states are removed by pruning.

In terms of sizes, the reduced model has compressed the requisite dynamics into 64 generalised goal states that are accessed via 986 generalised states, i.e., about half of the generalised states have been eliminated by pruning. The size of the largest likelihood matrix at the first level was 5 × 288 and the smallest was 5 × 1, noting that each pixel can be one of 5 states. The corresponding sizes of the transition tensors range from 288 × 288 × 32 (for the largest group of pixel modalities) to 2 × 2 × 2 for the reward modality. At the second level, the largest of 12 likelihood mappings—was 288 × 986. The transition tensor among the highest-level generalised states was 986 × 986 × 9. The question is now whether inductive inference can pick out paths of least action under this reduced model to produce expert play.

### 4.3. Phase 3: Continual Learning

Figure 8 reports the results of continual learning over 50 games of 200 frames each. Recall that continual learning accumulates generalised states that lie within basins of attraction. The model starts with 986 generalised states and quickly accumulates ~50 further states over the first few games. However, after about 20 games there are no further states to be discovered, saturating at 1037. As might be anticipated, the number of latent paths remains at 9 throughout (because there are no further degrees of freedom to induce new generalised paths).

Crucially, the implicit agent demonstrates perfect play from the onset of continual learning, with 11 rewards and no losses for every game (lower right panel). In this game, a maximum of 11 rewards can be secured within 200 frames of gameplay. In short, the attractor discovery phase of de novo learning provides a sufficiently dense orbit for inductive inference to take the shortest path among goals and thereby evince expert performance. Interestingly, this did not involve any learning in the conventional sense: expert play arises from inductive inference under the causal (attractor) discovery based on Bayesian model selection. However, there is a progressive increase in the path integral of the ELBO over frames: see lower left panel. This reflects an increase in confidence and predictive accuracy as Dirichlet counts are accumulated through experience. This accumulation increases the precision of transition tensors, leading to more precise predictive posteriors and resulting action. (Precision increases because of the small concentration parameter added to each transition tensor for continual learning. However, it also increases because only paths of least action are traversed, resolving uncertainty about the most likely path from any given generalised state). This speaks to an important aspect of performance: it is not deterministic. Policy selection is based upon inference under uncertainty. This uncertainty arises at the level of slightly imprecise transition priors but also at the level of action selection, where we used a high (512) precision parameter to select the next action based upon posterior predictive densities over the proprioceptive stream (c.f., sticky action in machine learning benchmarks).

Figure 9 shows an example of gameplay over 512 frames after continual learning. The upper panels report posterior beliefs over generalised states at the highest (second) level and the discovered transitions (left and right, respectively). These generate posterior predictive densities over the hidden causes of dynamics at the first level (shown in image format). The ensuing fluctuations in negative variational free energy are shown for the two levels (ELBO panel). The green dots correspond to rewarded outcomes. The lower panels illustrate the first few frames of gameplay and the last frame.

As intimated in the introduction, there are periods of play during which the variational free energy does not change, speaking to a gradient-free engagement with the world. However, there are occasional dips in the ELBO. These occur when the agent is uncertain about what is going to happen next. For example, after hitting the last row of targets, the ball is reset at a random location, that cannot be predicted precisely. This introduces uncertainty and a transient reduction in the ELBO.

These results were obtained prior to the last form of model reduction that conserves the mutual information between the generalised states and their consequences that underwrite inductive inference (namely, the reward, punishment and paddle position). The corresponding results following compression are shown on the right panels of Figure 9. Here, the periods of uncertainty—as scored by fluctuations in the ELBO—are now more frequent. This follows because the agent has now represented states of the world in terms of this OR that, which necessarily incurs ambiguity about state transitions. However, inductive inference can still maintain cost-free play. Interestingly, there were a couple of occasions in which the agent chose to hit non-rewarding targets to keep the ball in play: compare the green dots on the right and left of Figure 9.

This final stage of model reduction reduced the number of generalised states from about 1037 to ~700, i.e., by about 30%. This can be seen by comparing the size of the transition matrices before and after compression in Figure 9.

## 5. Conclusions

We have described a fast form of structure learning that enables generative models to assemble themselves on exposure to data. Sequences of observations are assimilated in terms of a compressed representation of dynamics. This enables the model to compose transitions, retaining only those that afford paths that intervene between certain (e.g., rewarded) states, while avoiding other (e.g., costly) states. With sufficient exposure and suitable (Bayesian) model reduction, one can, in principle, learn the structure and parameters of a generative model within a few minutes on a personal computer—and deploy such models as active inference agents to realise, reproduce or simulate Bayes-optimal behaviour. We close with some discussion of limitations and future directions.

One might ask whether de novo learning of this sort will always work. Clearly, the choice of the number of exposures—and size of basins of attraction—will depend upon the nature of the generative process (e.g., physics or game engine). For example, the number of exposures during attractor learning depends upon the minimal number of transitions necessary to support a pullback attractor (i.e., a transitive closure problem). However, this presents a chicken and egg problem; in the sense that transitive closure is the objective of de novo learning. This means, one has to try different schedules of attractor (and continual) learning—by increasing exposures until the model is sufficiently complex or expressive to support performant agency. For example, we repeated the above example but terminated attractor learning after a closed orbit among 32 goal states had been discovered, and tested performance prior to continual learning. Figure 10 shows a failure of inductive inference after about 200 frames of gameplay, with costs incurred by missing the ball (red dots). The agent is able to recover but still makes occasional mistakes. Note that mistakes elicit surprise, in the sense there is a profound drop in the ELBO (of 20 natural units or more). These surprises are generally resolved during continual learning, as the agent learns from its mistakes by accumulating the transitions it inferred were the best response. Again, the amount of continual learning depends upon the game in question.

A related question pertains to stochastic or random effects and how they are handled by this approach to de novo learning. It is helpful to distinguish between random effects such as noisy data or sticky actions on the one hand, and systematic uncertainty that is inherent to the cause-effect structure of the game, e.g., balls or bombs appearing in one of many locations. The former kind of uncertainty is generally resolved through the coarse graining implicit in renormalisation. This follows because irrelevant (random) fluctuations are discarded at each scale: c.f., [82]. For example, if the position of an object varies by a few centimetres at a fine scale, this variation has no effect at a scale that locates the object to within a metre. In the example above, we assumed that a potentially high dimensional pixel input has already been coarse grained to provide a summary of discrete transitions.

Indeterminacy after coarse graining is handled by assigning every possible transition to a new path, such that indeterminacy in the dynamics (e.g., random reset locations) is encoded in terms of predictive posteriors over paths. In the limit of very unpredictable transitions the size of the requisite transition tensors will increase and become incompressible. Put simply, this means that if there is too much randomness then there is no simple cause-effect structure that can be learned efficiently. In this setting, Bayesian model reduction may remove entire levels, furnishing a shallow agent that can only respond locally in space and time. This conjecture invites some interesting questions about how an agent should sample its sensorium most efficiently, i.e., maximising mutual information between its inferred latent variables and locally observable consequences.

### Limitations and Future Directions

There are a number of obvious limitations to de novo learning of this sort. The first inherits from the need to see all allowable transitions in the dataset. This speaks to a potential failure to generalise from sparse data in a way that continuous state space models can. This rests upon the fact that discrete state space models have no notion of metric space and implicit constraints on the movement of objects. In turn, this means there is no straightforward way to leverage prior knowledge about the (e.g., classical) mechanics generating observations. This can be contrasted with top-down schemes that have explicit representations of position, motion, forces, etc., e.g., [95]. One could argue that discrete state space representations are only apt when the process generating observations cannot be described in terms of classical mechanics; for example, language, pattern formation, and so on. On this argument, the latitude afforded by discrete state space representations may be useful, provided the numerics or degrees of freedom of the requisite models are sufficiently small. This brings us to the next limitation.

Discrete state spaces are flexible but costly in terms of their representation. In part, this inherits from the above limitation; in that it is not easy to place metric constraints on transition priors. For example, a single latent state—such as position—is now replaced with a factor whose states correspond to distinct positions. Although the discrete representation has the latitude to model any dynamics, it can put pressure on the numerics and the size of the problem to which de novo learning can be applied. A pushback against the lost opportunity for generalisation rests upon the opportunity to reduce the model, with no loss of mutual information between discrete (generalised) states and observable consequences. This could be regarded as a form of generalisation, in the sense that a combination of patterns (i.e., one combination of observations or another)—that are relevant for action—are retained.

In the introduction, we suggested that the learning here is entirely de novo, in the sense of requiring only data and no priors. While this is true in one sense, one could interpret the procedures we have described here as instantiating prior beliefs that are universal. For instance, we assume that (stochastic) limit cycles are more probable than point attractors (i.e., absorbing states), that some observations (or states) can be categorised as rewarding and others as costly, and that the best models of some data are those for which the mutual information between latent variables and data are maximal. However, these priors are properties of the sort of agent—or ‘strange particle’ [2]—that we are interested in. It may be possible to design, or find in nature, artefacts that violate these properties. Such artefacts would be static, fail to seek out goal states, or develop models lacking the sparse conditional dependency structures required to engage in inductive planning. For example, it is highly unlikely that de novo learning of this sort would ever learn to play chess, simply because chess has point attractors (i.e., checkmate).

The implicit constraints on the size of tensors is a limitation in the sense of the so-called scaling problem. Indeed, this note showcased a small problem due to these constraints. However, this constraint was not due to de novo learning of the generative model, which can handle large problems through the use of renormalising models. The constraint was imposed by the size of the tensors used to generate the outcomes, i.e., encoding the generative process. This follows because generative processes involving N objects, each with M quantised states and U joint paths, requires large B∈ℝN⋅M×N⋅M×U transition tensors. This follows from the fact that objects interact with each other and cannot be factorised into N smaller tensors. From a biomimetic perspective, this may be a fundamental constraint on dealing with high-dimensional sensory data. The argument here is that most phenotypes selectively sample the sensorium with a limited field of view, e.g., the foveal representation in human vision [96]. In short, by selectively sampling observation space, one can eschew the computational complexity of dealing with high-dimensional observations. In turn, this speaks to potentially interesting extensions of de novo learning, in which the observations are sub-sampled or supplied in a frame of reference that moves with the agent (or manipulanda), e.g., [97]. This and many other issues offer interesting avenues for exploring the limits or utility of de novo learning.

## Figures and Tables

**Figure 1 entropy-27-00992-f001:**
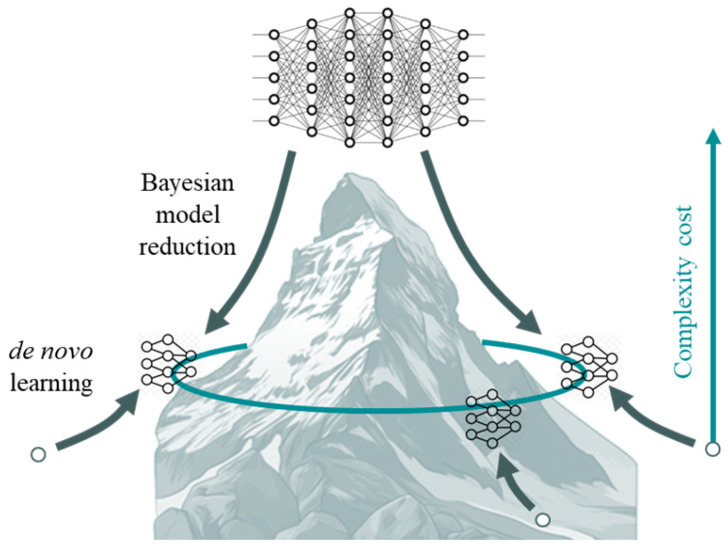
Model selection. The schematic illustrates the distinction between a top-down approach to optimising the size or structure of a neural network in relation to its complexity. One can either start with a large model or reduce it, to provide an efficient model of domain-specific data. Alternatively, one can start from nothing and grow the model to the requisite complexity. In neurobiology, it is likely that both bottom-up and top-down model selection play a role at evolutionary and neurodevelopmental scales. For example, most phenotypes are equipped with an overly expressive generative model that is progressively pruned and specialised with accumulating experience: see [40]. However, this accumulation and subsequent reduction may be recapitulated at many timescales; from the sleep–wake cycle [41], through to moments of reflection and introspection [42]. We will touch upon these dual aspects of continual structure learning in subsequent sections.

**Figure 4 entropy-27-00992-f004:**
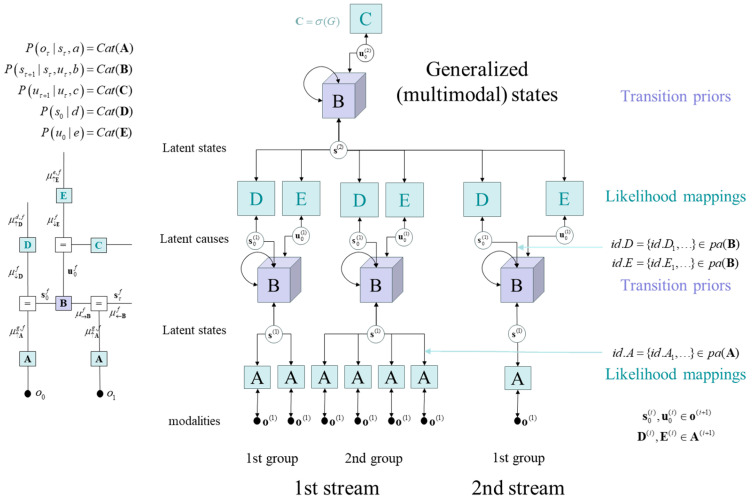
Renormalising generative model. The panel on the left reproduces a sub-graph of the factor graph in the previous figure. The schematic on the right illustrates the architecture of renormalising generative models, where time renormalisation has been omitted for clarity. In these models, the latent states at any given level generate the initial conditions and paths of [groups of] states at the lower level. This entails a separation of temporal scales, such that higher states only change after a fixed number of lower state transitions. This kind of model can be specified in terms of (i) transition tensors (B) at each level, encoding transition priors under each discrete path and (ii) likelihood mappings between levels, corresponding to the $D^{\left(i\right)},E^{\left(i\right)}\in A^{\left(i+1\right)}$ matrices that furnish empirical priors over the initial states (and paths) at level i. Because each state (and path) has only one parent, the Markov blanket of each state ensures conditional independence among latent states. In sum, a renormalising generative model is a hypergraph, in which the children of states at any level are the initial states and paths at the level below. In this example, there are only two levels, and the first level has been divided into two classes of modalities or streams. The streams are grouped together at the second level, such that outcomes in both streams are generated by transitions among the generalised states of a single factor. The edges have been equipped with arrows at both ends to indicate that messages are passed in both directions during model inversion (i.e., variational inference). In sum, the model can both generate and recognise outcomes.

**Figure 5 entropy-27-00992-f005:**
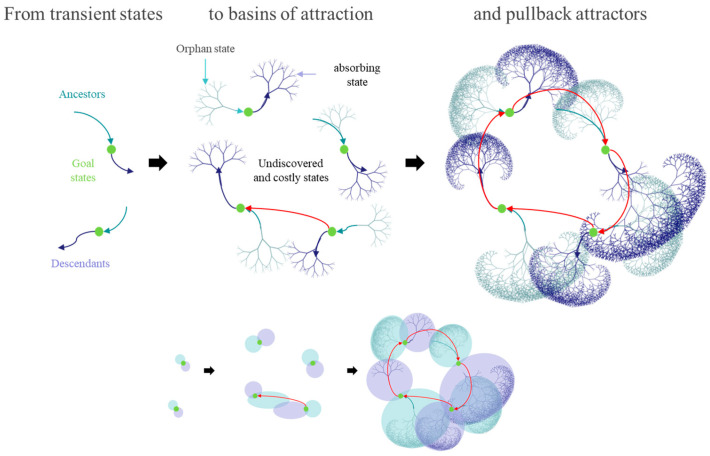
Topological transitivity. This schematic illustrates the growth of a model in terms of generalised (i.e., deep) states that are accumulated in the second phase of de novo learning. From left to right, we start with transient states that generate rewarding observations. Transitions among these states are retained if they are ancestors (i.e., predecessors) or descendants (i.e., successors) of goal states. This is illustrated in the lower panel in the form of a Venn diagram, where puce corresponds to ancestral states and purple the descendants (where goal states can be regarded as the intersection). Some ancestral states will not have predecessors (i.e., orphan states in the upper schematics). Similarly, some descendants will not have successors (i.e., absorbing states: strictly speaking, these are not absorbing states because—after normalisation of the transition tensor—there is a uniform probability that they will transition to every other state). With sufficient exposure, the forward and backwards basins of attraction (i.e., orbits) will grow. At some point, the descendants of one goal will intersect with the ancestors of another, inducing an orbit from one goal state to the next (red lines in the middle panels). A pullback attractor emerges when a subset of goal states can all be reached from each other. Clearly, the discovery of such an attracting set depends upon its existence; however, in principle, if an attracting set exists, it should be discovered in the limit of exposure to very long sequences. Recall that generalised states generate short paths in observation space and therefore attractors in generalised state-space refer to paths among paths. The final phase of de novo learning is to realise the shortest paths (of least action) on this attracting set or closed orbit. Finally, by construction, the attracting set cannot include undiscovered or removed states, such as costly states that generate punishing observations.

**Figure 6 entropy-27-00992-f006:**
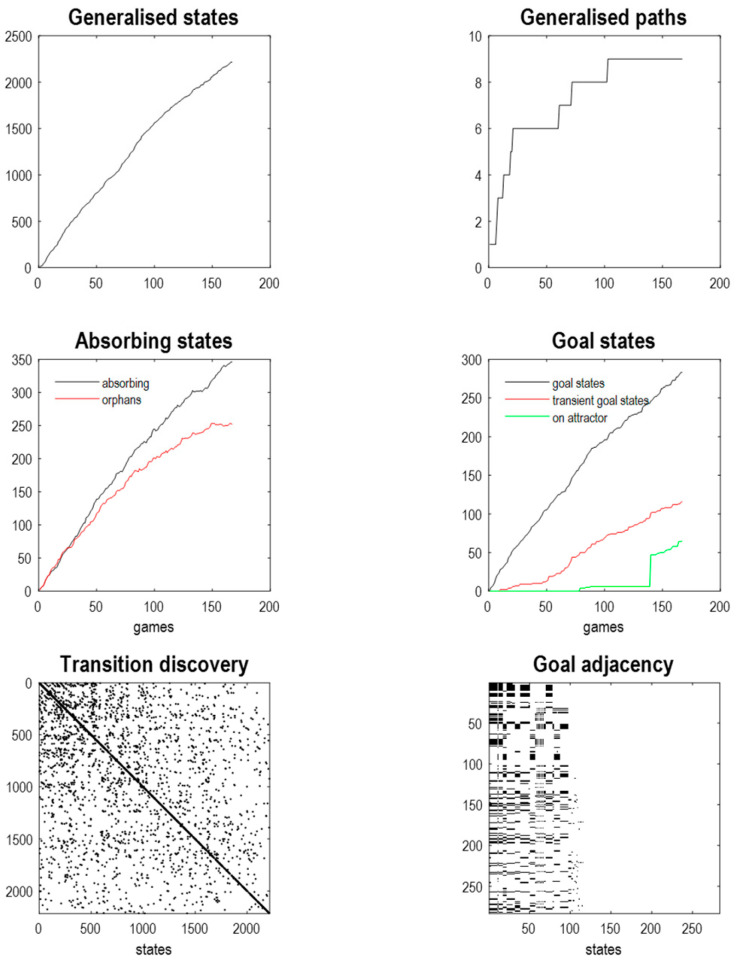
Attractor learning. This figure reports the learning of a pullback attractor over successive games or learning epochs. The upper panels show the number of generalised states and paths that are progressively discovered. The middle row reports on generalised states (left panel) and goal states (right panel). Generalised states are characterised in terms of the number of absorbing and orphan states, while goal states are enumerated in terms those that have successor goal states and those that have both successors and predecessors (i.e., goal states on the pullback attractor). The lower panels report transitions in terms of adjacency matrices for all generalised states (left) and goal states (right), respectively.

**Figure 7 entropy-27-00992-f007:**
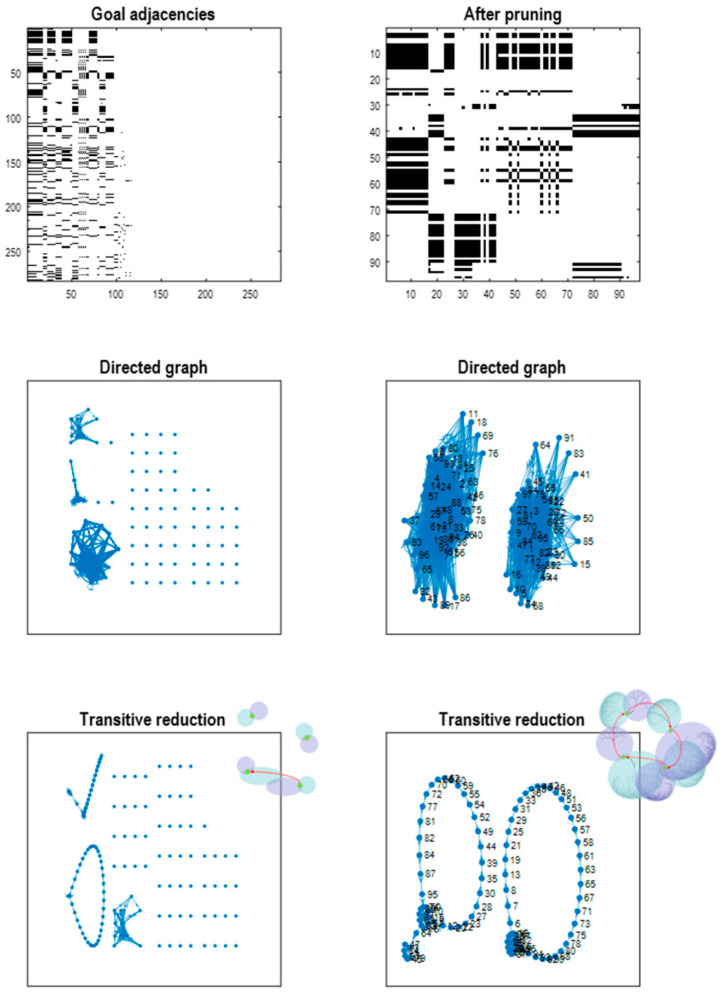
Transitive reduction. This figure illustrates the effect of pruning to reveal closed orbits or pullback attractors. The same characterisations are reported before (left column) and after (right column) pruning. The adjacency matrices (upper panels) are portrayed in terms of directed graphs before (middle panels) and after (lower panels) transitive reduction.

**Figure 8 entropy-27-00992-f008:**
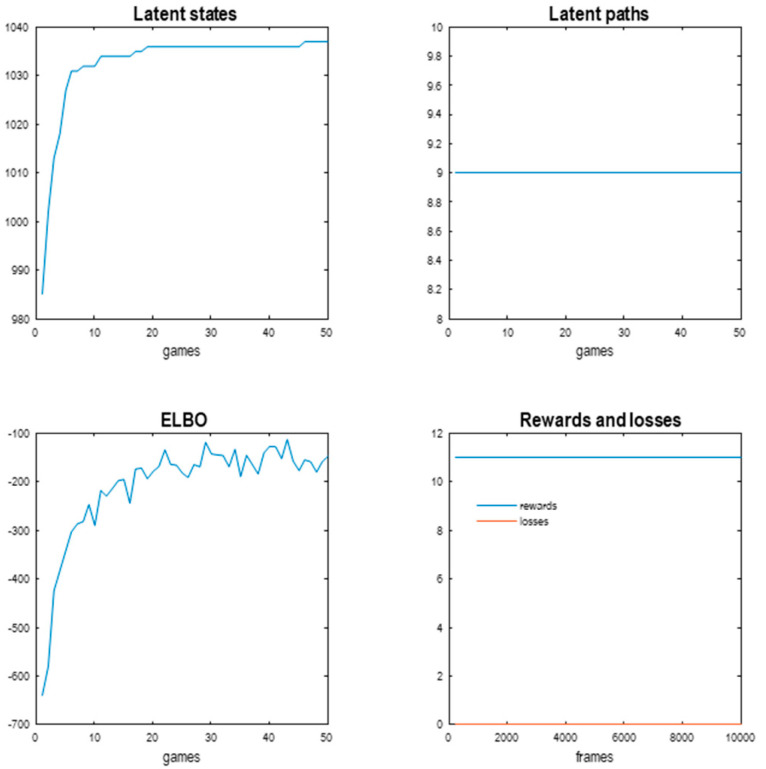
Continual learning. This figure reports continual learning in terms of the number of generalised states (upper left) and paths (upper right) accumulated over successive games. The lower panels report performance in terms of negative variational free energy (i.e., ELBO on the lower left) and the number of rewards and losses incurred for each game (lower right).

**Figure 9 entropy-27-00992-f009:**
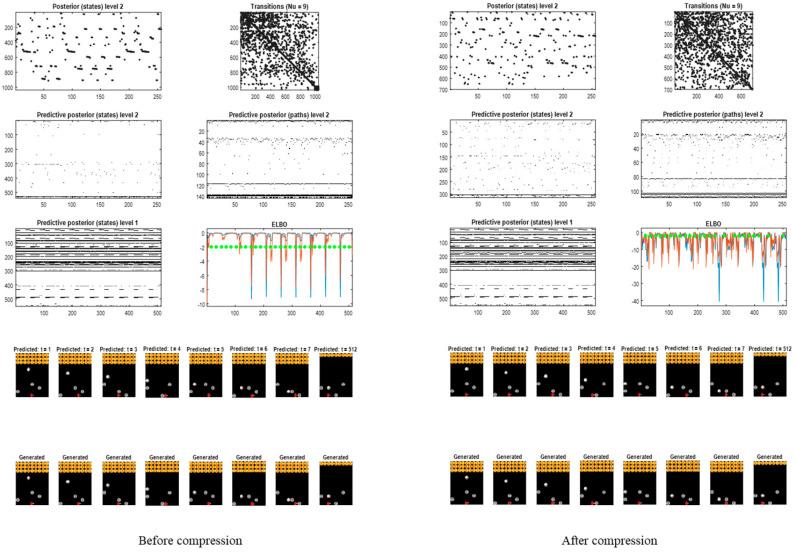
Inductive inference. This figure reports active inference under the model after 50 games of continual learning before (left)—and after (right)—further compression to merge generalised states while preserving mutual information with trailing streams. Each panel reports probabilities, with white representing zero and black one. The upper right panel shows the discovered transitions among (high level) generalised states. The upper left panel depicts the posterior distribution over states at the highest level in image format; here, 256 transitions. These latent states then provide empirical priors over initial states of groups at the first level, depicted in the predictive posterior panel below. The accompanying predictive posterior over paths are shown on the right; thereby generating predictive posteriors in pixel space. The initial and last images are shown in the lower row in terms of posterior predictions (upper row) and those generated by the game engine (lower row). The panel labelled ELBO reports fluctuations in the evidence lower bound over 512 frames of gameplay for the first and second levels (blue and red lines), along with time points that were rewarded (green dots).

**Figure 10 entropy-27-00992-f010:**
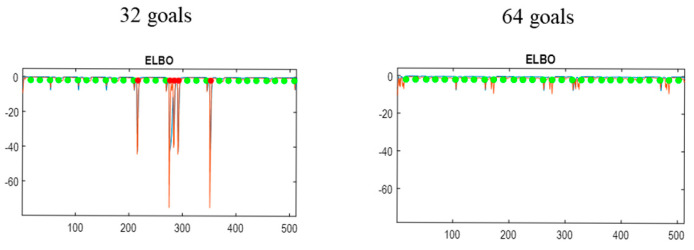
Making mistakes. This figure uses the same format as the ELBO panels in the previous figure. These results are based upon a model after attractor learning but no continual learning, where attractor learning was terminated after 32 (left panel) and 64 (right panel) goal states could be reached from each other. The green dots report rewarding episodes and the red dots indicate losses after the ball was missed.

## Data Availability

The data presented in this study may be available on request from the corresponding author.

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
