# Peer review of "Gradient-Free De Novo Learning"

_entropy, 2025, doi:10.3390/e27090992_

Round 1
Reviewer 1 Report
Comments and Suggestions for Authors
This is a great piece of work on de novo learning. The paper integrates and presents many concepts, which requires specific knowledge in order to understand it fully. Below are some of my humble comments for clarification:
- In the abstract, the authors used two sentences to mention the Bellman optimality principle. While this paper mainly employs the Free Energy Principle — and I understand that the distinction between the Free Energy Principle and Bellman optimality principle is conceptually important — but these are not the key points of de novo learning. Why not instead use two sentences to directly describe the key characteristics or core ideas of de novo learning?
- When equipped with hierarchical depth in the POMDP (as shown in Figure 2), are A, B, D, and E the same sets at the high level as they are at the low level?
- On page 10, in Equation (6), what is the difference between bold a and regular a?
- In line 349, should "Figure 2" actually be "Figure 3"?
- For “a worked example,” I would suggest adding illustrative figures for the games — also showing how to choose the hidden states, parameters, actions, etc.
Author Response
Dear Reviewer,
Thank you very much for your comments. Please find a point-by-point response in the attached document under Reviewer 1.
With kind regards,
Lancelot Da Costa

Reviewer 2 Report
Comments and Suggestions for Authors
The paper “Gradient-free de novo learning” by K. Friston et al. addresses the challenging task of developing a novel formalism for reinforcement learning based on active inference combined with the free-energy principle. Written by leading specialists in the active inference approach to cognitive phenomena, it presents original ideas in this field and undoubtedly merits publication in Entropy.
However, before publication, I recommend that the authors revise the manuscript to take into account the following:
1. Concerning the Introduction:
It contains many overly general propositions that, on the one hand, are unnecessary for the issue under consideration and, on the other hand, have counterexamples. For instance:
Lines 39-41: “the free energy principle allows one to describe any open system—i.e., a system that senses and acts upon its environment—as inferring the causes of its observations and consequent actions.”
Such systems cannot be classified simply as general open systems. For example, non-equilibrium reaction–diffusion systems are also open systems in which various emergent phenomena have been identified and studied in detail. However, for these systems no general principles—comparable to free-energy minimization—have yet been established (e.g., B. S. Kerner, V. V. Osipov, Autosolitons: A New Approach to Problems of Self-Organization and Turbulence, Springer, 1994).
Lines 58-59: “First principles accounts of self-organisation generally call upon variational principles of least action.”
A simple counterexample is self-oscillations emerging via a subcritical bifurcation characterized by strong dissipation. I think the general principle of self-organization—the emergence of spatiotemporal patterns—is the loss of stability of the structureless state of the system in question. The development of instability can be described using variational principles only for certain types of systems.
Lines 70-73: “By the Helmholtz-Hodge decomposition, this flow has dissipative and conservative parts (Evans, 2003; Kwon et al., 2005; Seifert, 2012). In other words, the flow or dynamics can be separated into a dissipative (curl-free gradient flow) and a conservative (divergence-free solenoidal flow) part that circulates on the isocontours of variational free energy.”
For turbulent motion characterized by multiscale spatiotemporal patterns of vertices or general discrete Markov systems, this type of decomposition is problematic.
The given examples merely illustrate that a more specific description of such systems is required. In fact, the text notes that such a system “maintains itself within certain characteristic states and is able to separate itself from the environment.” It is necessary to emphasize this feature more explicitly.
The text in lines 89–105, which contrasts reinforcement learning and Bayes optimality by distinguishing functions of states from functions of beliefs about states, is not easy to understand. In a relatively simple model-free approach to reinforcement learning, rewards—treated as functions of states—are accumulated in option preferences, which are functions of beliefs about states and thus subjective quantities. This accumulation reflects human characteristics, such as the overestimation of rare events and the underestimation of frequent events. Therefore, it would be desirable to elucidate the principal differences between reinforcement learning that incorporates these human aspects and learning algorithms based on Bayesian optimality that also reflect cognitive factors.
The subsection “Model Selection” addresses a general and important issue in the theory of self-organization, specifically the construction of meso-level models based on both top-down and bottom-up approaches. In this context, it is desirable to highlight the general concept of collective variables, which underpins a wide range of theories describing emergent phenomena in systems of various kinds, from superconductivity to brain functioning. It is essential that such meso-level variables govern system dynamics at the macro-level while also “enslaving” micro-level variables in a quasi-stationary manner. Regarding brain functioning, the concept of collective variables has been elaborated by Kelso (e.g., Kelso, J. A. S. & Tognoli, “Toward a Complementary Neuroscience: Metastable Coordination Dynamics of the Brain,” in E. Murphy, N.; Ellis, G. & O'Connor, T. (Eds.), Downward Causation and the Neurobiology of Free Will, Springer-Verlag, 2009, pp. 103–124).
In the subsection “Rewards and Punishment,” the authors distinguish between punishment as a constraint and as a negative reward. However, if an agent initially has no information about a possible punishment when selecting a strategy to achieve its goal, how can the agent avoid it? The agent simply selects this strategy, receives a negative “reward,” and subsequently avoids selecting it again. In this case, what is the difference between constraints and rewards? Please clarify.
2. Concerning the Active Inference, Learning, and Selection:
The subsection “Generative Models” introduces the quantities underlying the subsequent constructions. For example, P(o∣s,a) denotes the probability of observing o when the system is in state s and the agent takes action a ??; and P(s′∣s,u,b) denotes the probability of the system transitioning to state s′ at the next time step, given that it is currently in state s, the selected action strategy is u, and the expected system properties (beliefs) are b. Unfortunately, these quantities are presented only in the caption of Fig. 2. I strongly recommend that they be specified in the main text as a clearly organized list, with a detailed explanation of each argument used. The same applies to the Dirichlet distribution with its counts as arguments (and the Categorical distribution), which could perhaps be supplemented with a brief appendix emphasizing their basic features and utility in statistics.
The same applies to the subsections “Variational Free Energy and Inference” and “Active Inference”: please first introduce and explain the quantities used, and then specify the free energy. Figures are usually used to illustrate the text, and the use of their captions to introduce the basic quantities is not convenient for readers.
3. Concerning A Worked Example:
The main sections — Active Inference, Learning and Selection, and Attractor Learning — are written using a rather general formalism, the details of which remain unclear without reference to the authors’ previous publications. Therefore, the section A Worked Example is particularly important for elucidating the presented formalism. Unfortunately, the current text explains only the underlying concepts rather than the formalism itself. In this context, I recommend that the authors extend this section to illustrate the specific form the main formulas take in the analyzed example.
4. The present paper puts forward a novel approach to learning that may be regarded as a potential competitor to reinforcement learning. Therefore, it could be useful if the authors also would considere a simple example in which the results of a reinforcement learning algorithm are well known — for instance, the rock–paper–scissors game (e.g., L. BuÅŸoniu, R. Babuška, B. De Schutter, IEEE Trans. Syst. Man Cybern. Part C Appl. Rev. 38, 156 (2008); Y. Sato, J. P. Crutchfield, Phys. Rev. E 67, 015206 (2003); Y. Sato, E. Akiyama, J. P. Crutchfield, Physica D 210, 21 (2005)). The Nash equilibrium in this game corresponds to the situation where both participants select their options randomly and with equal probability.
Let us assume that one of the participants does not behave optimally — for example, his action selection is biased, shows correlations between consecutive time steps, or depends on the opponent’s current action. In such a case, it would be interesting to demonstrate the authors’ formalism by constructing a winning strategy for this game.
Author Response
Dear Reviewer,
Thank you very much for your comments. Please find a point-by-point response in the attached document under Reviewer 2.
With kind regards,
Lancelot Da Costa

Round 2
Reviewer 2 Report
Comments and Suggestions for Authors
The revisions satisfactorily address the referee’s concerns, and I now support the manuscript’s publication.
Author Response
Dear Reviewer,
Thank you very much for your clear and constructive feedback that helped us improve the manuscript.
On behalf of all authors,
Kind regards,
Lancelot Da Costa